# Evidence of disorientation towards immunization on online social media after contrasting political communication on vaccines. Results from an analysis of Twitter data in Italy

**Samantha Ajovalasit**[1,2]*, **Veronica Maria Dorgali**[3], **Angelo Mazza**[1], **Alberto d'Onofrio**[4,5], **Piero Manfredi**[6]

**1** Department of Economics and Business, University of Catania, Catania, Italy, **2** Department of Environmental Science, Informatics, and Statistics, Ca' Foscari University of Venice, Venice, Italy, **3** Department of Statistics, Computer Science, Applications "G. Parenti" (DISIA), University of Florence, Florence, Italy, **4** Department of Mathematics and Statistics, University of Strathclyde, Glasgow, United Kingdom, **5** International Prevention Research Institute, Lyon, France, **6** Department of Economics and Management, University of Pisa, Pisa, Italy

* samantha.ajovalasit@unive.it

**Data Availability Statement:** Due to restrictions in the Twitter terms of service (https://twitter.com/en/

## Abstract

### Background

In Italy, in recent years, vaccination coverage for key immunizations as MMR has been declining to worryingly low levels, with large measles outbreaks. As a response in 2017, the Italian government expanded the number of mandatory immunizations introducing penalties to unvaccinated children's families. During the 2018 general elections campaign, immunization policy entered the political debate with the government in-charge blaming oppositions for fuelling vaccine scepticism. A new government (formerly in the opposition) established in 2018 temporarily relaxed penalties and announced the introduction of forms of flexibility.

### Objectives and methods

First, we supplied a definition of disorientation, as the "lack of well-established and resilient opinions among individuals, therefore causing them to change their positions as a consequence of sufficient external perturbations". Second, procedures for testing for the presence of both short and longer-term collective disorientation in Twitter signals were proposed. Third, a sentiment analysis on tweets posted in Italian during 2018 on immunization topics, and related polarity evaluations, were used to investigate whether the contrasting announcements at the highest political level might have originated disorientation amongst the Italian public.

### Results

Vaccine-relevant tweeters' interactions peaked in response to main political events. Out of retained tweets, 70.0% resulted favourable to vaccination, 16.4% unfavourable, and 13.6%

tos) and the Twitter developer policy (https://developer.twitter.com/en/developer-terms/agreement-and-policy.html) we cannot provide the full text of tweets used in this study. However, for replication purposes, we provide the ID of every tweet used, also containing the manually labeled training set adopted; details on how to fetch tweets given their IDs are provided in https://developer.twitter.com/en/docs/tweets/post-and-engage/api-reference/get-statuses-lookup. The dataset of the Tweets IDs corresponds to the data available up to January 7 2019. Furthermore, note that any sample of tweets containing the same set of keywords listed in the manuscript and posted over the same time period is likely to yield study findings similar to those reported in the article. We extracted data available at that time from the Twitter public web interface; this data can also be purchased from Twitter via their Historical PowerTrack API http://support.gnip.com/apis/historical_api2.0/. The authors provide set of minumum requirements in form of Tweets IDs.

**Funding:** The authors received no specific funding for this work.

**Competing interests:** The authors have declared that no competing interests exist.

undecided, respectively. The smoothed time series of polarity proportions exhibit frequent large changes in the favourable proportion, superimposed to a clear up-and-down trend synchronized with the switch between governments in Spring 2018, suggesting evidence of disorientation among the public.

## Conclusions

The reported evidence of disorientation for opinions expressed in online social media shows that critical health topics, such as vaccination, should never be used to achieve political consensus. This is worsened by the lack of a strong Italian institutional presence on Twitter, calling for efforts to contrast misinformation and the ensuing spread of hesitancy. It remains to be seen how this disorientation will impact future parents' vaccination decisions.

## Introduction

The dramatic success of immunization programs in industrialized countries, with decades of high vaccine uptake and ensuing herd immunity, is suffering a drawback, namely the generalized fall of perceived risks arising from vaccine-preventable infectious diseases. This promotes the spread of resistance, or reluctance, to vaccination. This phenomenon, nowadays identified as "vaccine hesitancy" [1–3] is currently considered one of the top threats to global health because of its pervasive and complex nature [4]. Ensuring vaccination programs' resilience to the hesitancy threat is a significant task of current Public Health systems.

In Italy, the MMR (measles, mumps, and rubella) vaccination coverage at 24 months, which was in the region of 91% in 2010, fell at 85.3% in 2015 and remained low after that. Parallel to this, large measles outbreaks, with 844 cases in 2016, 4,991 in 2017 (with four deaths), and 2,029 cases in the first six months of 2018 [5–7] were observed.

As a response, in the Italian National immunization plan for 2017–2019, the Italian government acted to increase the number of mandatory immunizations [8–11] by introducing penalties for non-vaccinators in the form of fines and restrictions to admittance to kindergarten and school. The decree's ethical implications, mainly the introduction of sanctions, have been strongly challenged, especially in online social media (OSM). With the 2018 general elections, the vaccination policy flooded the political debate, with the government accusing the opposition of fuelling scepticism around vaccination. The new government, established in June 2018 and composed by a coalition between an anti-establishment movement and a far-right party, allowed, after several contrasting announcements, unvaccinated children to be admitted to school.

Over the past fifteen years, OSM emerged as a major popular source of information, including health topics [12–14]. However, within OSM, anyone can express her/his own opinion, regardless of her/his expertise in the particular topic considered. As a result, parents' immunization decisions could be influenced by misconceptions and false information [15–17]. The massive misinformation pervading the OSM environment has been defined by the World Economic Forum as one of the main threats to current societies [15–19], in particular, because of the emergence of echo chambers, i.e., "polarised groups of like-minded people who keep framing and reinforcing a shared narrative" [16].

Although opposition to vaccination, favoured by equally misinformation, existed since the very introduction of the smallpox vaccine [20], recently, because of the increase in Internet

access and the birth of the new communication platforms, misinformation is spreading at unprecedented rates [16, 21].

We focused our analysis on Twitter, a microblogging service, which is considered, as well as Facebook, a public square where anyone can express and share opinions and participate in discussions. On Twitter, user A may see user Bs' messages without being involved in a direct relationship ("follow"). Twitter thus represents a social network and an information network at the same time. This makes Twitter different from Facebook because of its structure. For example, Facebook allows easier identification of echo chambers and homophily [22, 23].

For epidemiological purposes, Twitter data have been used for surveillance and descriptive studies, e.g., the spread of seasonal flu, the 2009 H1N1 pandemic outbreak, and the 2014 Western Africa Ebola outbreak. In all these examples, a clear correlation between the temporal spread of infections and social media interactions emerged [24].

Supported by the steadily increasing internet access, Twitter is currently one of the primary tools used by political leaders to communicate with their public [25–27]. However, this implies that when political leaders intervene on scientific subjects, such as immunization, they exert tremendous pressure on public opinion [28]. When health-related topics are the subject of political disputes, with contrasting information being massively delivered by not formally qualified persons, some individuals may be induced to change their opinions compulsively, originating a condition of disorientation. Properly defining disorientation and testing for its presence in Twitter signals is a main task of this article. A preliminary literature search on the subject suggested that the issue of "disorientation", though ubiquitously present in many disciplines such as medical and cognitive sciences, spatial and information sciences, and social science [29–31], does not seem to have received systematic attention in the literature on information, opinions and online social media. Generally speaking, "disorientation" can be simply a consequence of the lack of adequate information, of the over-exposition to information, including misinformation, and more generally, of information disorder [32]. All these factors can make it difficult for people to filter the masses of available information properly. To simplify things and develop simple tests for the presence of disorientation in data, we assumed that disorientation (towards vaccines) could be coarsely identified as the lack of well-established and resilient opinions among individuals, therefore causing individuals to change their opinions as a consequence of sufficient external perturbations. The question then shifts on which the perturbations might be "sufficient". Clearly, some perturbations–typically those arising as direct resp onses of the public to media news—can be very short-lasting. In relation to this, we define a concept of "short-term disorientation" as a state in which people keep changing suddenly (and often) their opinion on the debated subject because of the overwhelming impact of multiple contrasting information. However, other perturbations, such as those following from non-scientific arguments persistently promoted or supported at the highest political level, e.g., a political party, or even a government, might generate longer-term effects that we term here as "long-term disorientation".

Consistently, in this article, we used sentiment analysis to describe the trend in communication about vaccines on Twitter in Italy throughout 2018 and to evaluate polarity in the opinions about immunization as preliminary steps to bring evidence–by appropriate statistical tests—that the prolonged phase of contrasting political announcements on a sensitive topic such as mass immunization might have originated a condition of disorientation among the Italian public.

## Materials and methods

Twitter is an online social media and micro-blogging service born in 2006. Users ("tweeters") write texts ("tweets") of 280 characters maximum length, which are publicly visible by default

until users decide to protect their tweets. According to statista.com (accessed on March 13[th], 2021), in 2021, Twitter has 340 million (estimated) active users worldwide.

## Data extraction, transformation, and cleaning

We collected tweets in Italian containing at least one of a set of keywords related to vaccination behaviour and vaccine-preventable infectious diseases posted in 2018, using the Twitter Advanced Search Tool. In total, we retrieved 443,167 tweets. Keywords were chosen from a review of previous literature, and they were appropriately expanded for our purposes. Subsequently, we applied supervised classification techniques to screen out irrelevant tweets and analyze the polarity proportions of the retained ones. Consistently, we deliberately chose a broader set of keywords in order to retrieve the largest possible set of tweets and then apply finer tools to identify and leave-out noise.

Data cleaning was performed using the Python programming language. A probabilistic approach was used to re-filter tweets written in Italian; then, possible duplications were removed using the Tweets ID field with 318,371 tweets retained for the analysis. For each post, we tracked subsequent interactions by counting the number of re-tweets and likes.

## Tweets classification, sentiment analysis, and training set

Sentiment analysis deals with the computational treatment of opinions, sentiments, and subjectivity within texts [33, 34]. Here, we used sentiment analysis methods for classifying tweets. In our analysis, we identified four categories: (i) favourable (F), if the tweet unambiguously showed a convinced pro-vaccine position, (ii) contrary (C), if the tweet unambiguously showed a position contrary to vaccination, (iii) undecided (U), if the tweet was neither favourable nor unfavourable, (iv)out of context (OOC), if the tweet was unrelated to immunization or if it did not fit any of the preceding categories (e.g., if it was merely spreading news or linking to another source, without expressing an opinion or a clear position). Tweets from the latter category were removed from subsequent analyses. Throughout the rest of the article, we will generically label the observed proportions of the three categories (F, C, U) (computed over any time period) as the "polarity" proportions. The sum of the contrary and undecided proportions can be taken as an estimate of the hesitant proportion in the overall Twitter population during the period considered. Notably, this is a wide population, potentially including people not involved in vaccination decisions, neither currently nor in the future, and therefore not necessarily relevant for the future vaccine coverage. Nonetheless, they represent a large population participating in a hot public debate and, therefore, relevant to opinion formation.

A supervised classification procedure [35, 36] was used to classify tweets into the four categories previously defined. First, a training set was created by manually tagging a random sample of 15,000 tweets out of the 318,371 retained for the analysis. Manual labelling was done by 15 trained university students. In particular, 15% of these 15,000 tweets were intentionally duplicated to measure the mutual (dis)agreement among annotators. The resulting accuracy was 0.6298 (CI 0.6034–0.6557), with a 'Fleiss' Kappa of 0.410, resulting in a fair agreement.

Next, we manually reviewed the duplicated tweets and those that showed invalid content (such as hashtag only or URL only tweets). Tweets labelled during previous explorative analysis were added. Eventually, we obtained a set of 14306 unique tweets that made the training set. In the classification process, we used unigram and bigram; we kept the hashtag (#vaccino) and removed the mentions (e.g. @screenname).

Eventually, the training set was used to compare five alternative classification models based on the following algorithms: Classification Tree, Random Forest, Naive Bayes, Support Vector Machine (SVM), and K-Nearest Neighbors.

**Seeking evidence of disorientation in Twitter data.** Consistently with the proposed definition of disorientation, in what follows, we propose a few procedures aimed to test for the presence of disorientation about vaccination amongst tweeters in Italy. We distinguish between short- and long-term disorientation. The former deals with a condition in which people keep changing suddenly (and often) their opinion on the debated subject because of the overwhelming impact of short-term information disorder. The latter deals with longer-term opinion perturbations, as it can be the case when the highest political actors, e.g., a political party, or even a government, persistently promote or support non-scientific arguments, including forms of denialism, thereby generating longer-term disorientation effects, including disorientation waves.

**Short-term disorientation.** To seek short-term disorientation symptoms in the data, we applied a number of tests relying on the size of the deviations (measured through the variance) from appropriately defined average opinions. The tests were conducted considering all the tweets retained, assuming they represented a random sample of an appropriate underlying super population. In particular, we proposed three different tests.

## A basic multinomial test of daily tweeting trends

We applied a simple multinomial test to identify those changes in the polarity proportions resulting from randomness and separate them from those that did not. Our null hypothesis is that the (true) proportions of categories (F, C, U) were the ones observed throughout the entire year. In practice, we computed, for every day, the probability value (p-value) that the observed vector of opinion proportions is a (random) sample drawn from a multinomial population whose parameter vector is given by the overall yearly mean of the polarity proportions.

**A "running" multinomial test for fast-changing opinions.** To further understand the short term changes in opinions, we tested whether each observed daily vector of polarity proportions represented a random sample drawn from a "running" multinomial population whose parameter vector is given by the average polarity proportions observed over the preceding 15 days. The figure of 15 days, representing our null hypothesis, was selected somewhat arbitrarily as a minimal duration representing a "stable" opinion (or "average preferences persistence") in the short term. However, a sensitivity analysis was conducted to check the robustness of this choice.

**A running-variance test.** Furthermore, all along the observed period, we computed a running 15-days variance of the proportion favourable to vaccination and tested (by the standard Chi-square) the null hypothesis that the 15-days variance is equal to the overall variance throughout the entire year.

**Longer-term disorientation.** As for long-term perturbations, we applied a polynomial fit to the smoothed polarity proportions trend over the entire year to look for possible evidence of long-term disorientation amongst the public. Smoothing was carried out using a discrete beta-kernel based procedure proposed by [37]; the use of beta kernels allows overcoming the problem of boundary bias, commonly arising from the use of symmetric kernels. The finite support of the beta kernel function can match our time interval so that, when smoothing is made near the time interval boundaries, no weight is allocated outside the support. The smoothing bandwidth parameter has been chosen using cross-validation.

## Results

### Automatic data classification and polarity proportions

Among the five classification algorithms tested, the Support Vector Machine (SVM) performed best (details in the online appendix), and it was consequently adopted. Main summary

**Table 1. Results of the support vector classifier (the classifier eventually selected) for the four categories considered in this work (favourable, contrary, undecided and out of context).**

|  | precision | recall | f1-score | support |
|---|---|---|---|---|
| Favorable | 0.43 | 0.46 | *0.44* | 785 |
| Contrary | 0.24 | 0.19 | *0.21* | 318 |
| Undecided | 0.20 | 0.15 | *0.17* | 299 |
| Out of Context | 0.63 | 0.67 | *0.65* | 1460 |
| Accuracy |  |  | *0.50* | 2862 |
| Macro avg | 0.37 | 0.37 | *0.37* | 2862 |
| Weighted avg | 0.49 | 0.50 | *0.49* | 2862 |

results based on standard measures [38] are reported in Table 1. These measures are consistent with the classification provided by human annotation (S5 Table reported in the S1 File). As mentioned in the previous section, by selecting a broad set of keywords, we chose to retrieve a larger set of tweets and left to supervised classification algorithms the task of identifying noise. Consistently, 57.8% of the total tweets were classified as out-of-context and discarded. Of the remaining tweets, the overall proportions of classified as favorable, contrary and undecided were: F = 70.0% (CI: 61.5–74.0), C = 16.4% (CI: 12.7–25.2), U = 13.6% (CI: 8.06–20.5), respectively.

**Hesitancy.** The proportion of hesitant individuals in our overall Twitter population, given by the sum of the contrary and undecided proportions, resulted in 30,1%.

**Institutional presence on Twitter.** The Italian Ministry of Health use of Twitter is relegated to press communications and the16.4 publication of statistics. Between 2013 and September 18th, 2019, the Italian Ministry of Health tweeted 2,454 times (of which 172 included the word vaccin*), i.e., 25% the figure observed in France from the Ministère des Solidarités et de la Santé. Essentially the same holds for the Italian National Institute of Health.

**Temporal trends.** The daily levels of Twitter interaction (including original tweets and subsequent likes or re-tweets) for 2018 (see Fig 1) show three prominent peaks, each accounting for hundreds of thousands of interactions. These three peaks represent users' responses to well-identified triggering events. The first peak, recorded on June 22nd, 2018, is the second-highest; it follows an Italian Minister of Interior public speech that defined the number of mandatory immunizations in the National Immunization Plan as "intolerably excessive". Polarity proportions observed on this day were F = 71.7%, C = 15.8%, and U = 12.5%, respectively. The second peak, recorded on August 4th, 2018, is the highest; it follows a government decree that suspends sanctions, such as non-admission to school, imposed by the previous government on unvaccinated children. Notably, whereas the number of tweets on this day exhibited a dramatic increase compared to the previous days, the underlying polarity proportions (F = 77.7%, C = 11.8%, and U = 10.5%) showed only moderate variations. The third peak (September 5th, 2018) follows the government's change of position on easing the sanctions on unvaccinated children (F = 73.6%, C = 14.4%, U = 12.0%). The graph in Fig 1shows a number of further lower peaks, still attributable to interventions in the political debate, over a long-term background of low-level activity.

## Testing the presence of disorientation

With the caveats reported above, the proportion of people "not favourable" to immunization–around 30%—was a worrying symptom of the complicated state of opinions about vaccination in Italy. The results of the various procedures proposed to investigate disorientation are reported below.

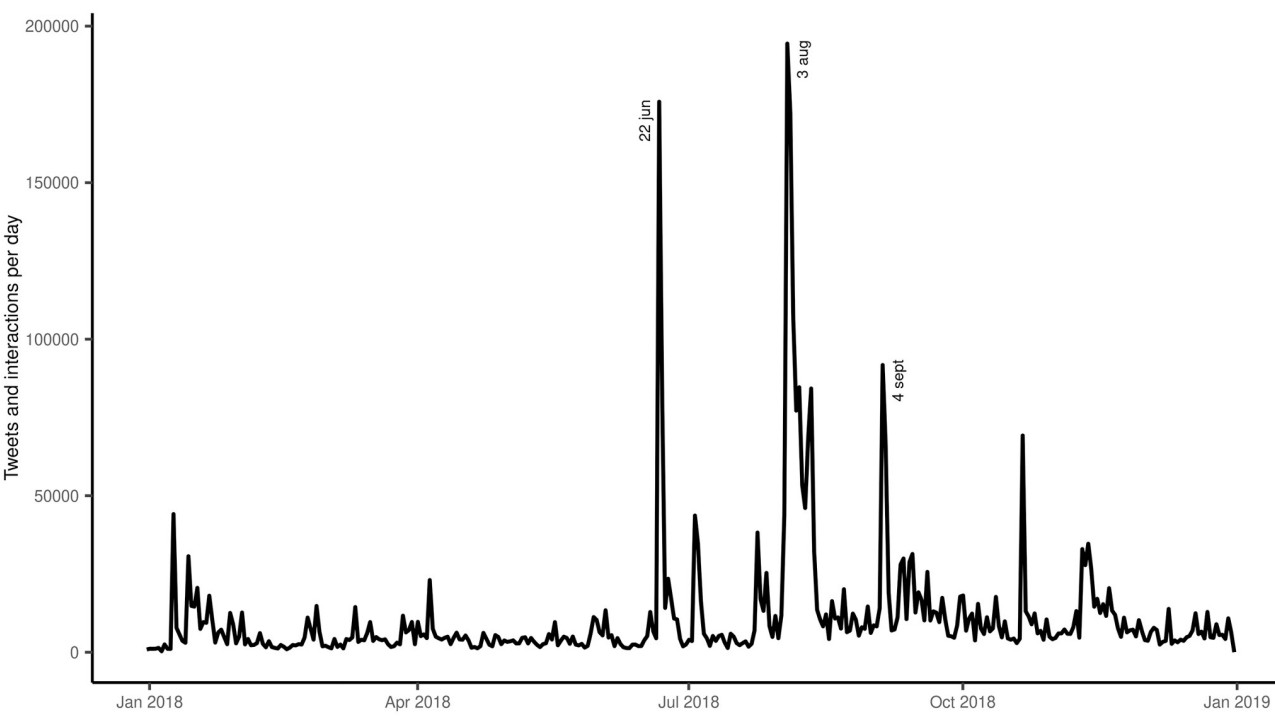

**Fig 1. Tweeting about vaccines in Italy during 2018: Time series of total daily interaction counts (tweets, likes, and re-tweets) and exact dates at main triggering political events or speeches.**

**Short-term disorientation.** Using as null hypothesis the polarity proportions observed for the whole year (F = 70.0%, C = 16.4%, U = 13.6%), the basic multinomial test (Fig 2) is significant in 132 days ($\alpha$ = 5%), against an expectation of approximately 18 days (5% of 365).

The running multinomial test (Fig 3) is significant in 101 days ($\alpha$ = 5%), providing further evidence of instability in polarity proportions.

Last, the running-variance test (Fig 4) is significant in 80 days ($\alpha$ = 5%). In particular, significantly high variances appeared in February and March 2018, at the end of the electoral campaign, and around the voting days. In contrast, significantly low variances appeared after the new government took office and before schools opening, suggesting a possible stabilization of opinions after the transition from one government to the next.

Overall, the three tests performed agree in bringing statistical evidence towards a rapid shift in vaccination opinions, denoting the presence of short-term disorientation according to the first definition provided.

**Smoothing and longer-term disorientation.** The smoothed time series show that many of the sudden changes of the daily polarity proportions originate from a rather small number of more stable and longer-lasting fluctuations (Fig 5). For the proportion favourable to immunization, the amplitude of these more stable oscillations is remarkable (from 60% to 76%), suggesting a substantial size of the "non-resilient" component of the population favourable to vaccination.

A stepwise polynomial fit of the smoothed polarity proportions (Fig 5) selected the parabolic function as the best one, allowing for a dramatic increase in the determination coefficient compared to the linear case, whereas higher-order functions increased $R^2$ only negligibly ($R^2$ values are reported in Fig 5).

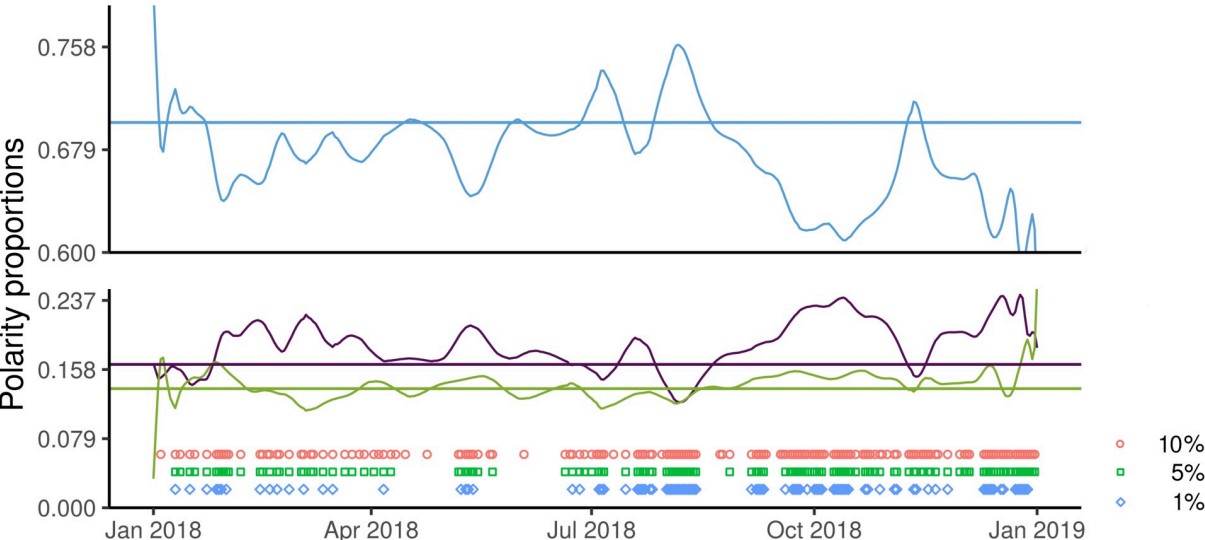

**Fig 2. Results of the basic multinomial test.** Blue circles, green squares, and purple diamonds denote the days when the null hypothesis was rejected at the significance levels of 10%, 5%, and 1%, respectively. For readability, we showed the smoothed polarity proportions. In the online appendix, we have reported in S1 Fig the **real (raw)** proportions used in the multinomial tests.

Between January and May, the parabolic trend exhibits a clear increase of the favourable proportion (and a parallel decline in the proportions undecided and contrary), possibly reflecting the "tail" of the positive impact of the "vaccine decree" issued by the previous government, and a marked decline thereafter, when the new government had taken office, losing more than 7% by the end of the year. While we are not able to provide a direct causal link between the

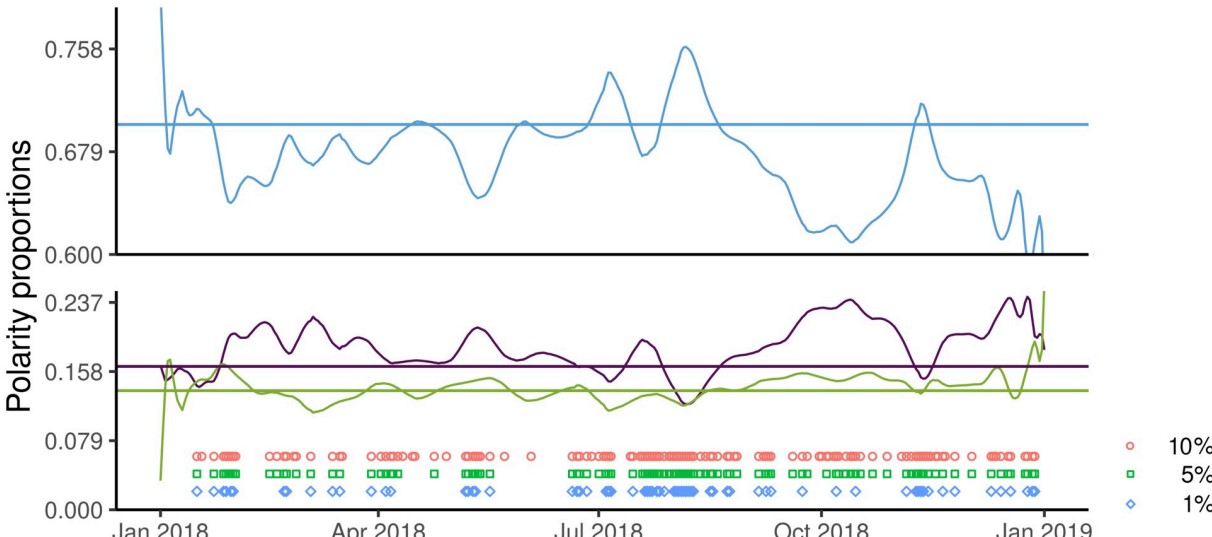

**Fig 3. Results of the running multinomial test at 15 days.** Blue circles, green squares, and purple diamonds denote the days when the null hypothesis was rejected at the significance levels of 10%, 5%, and 1%, respectively.

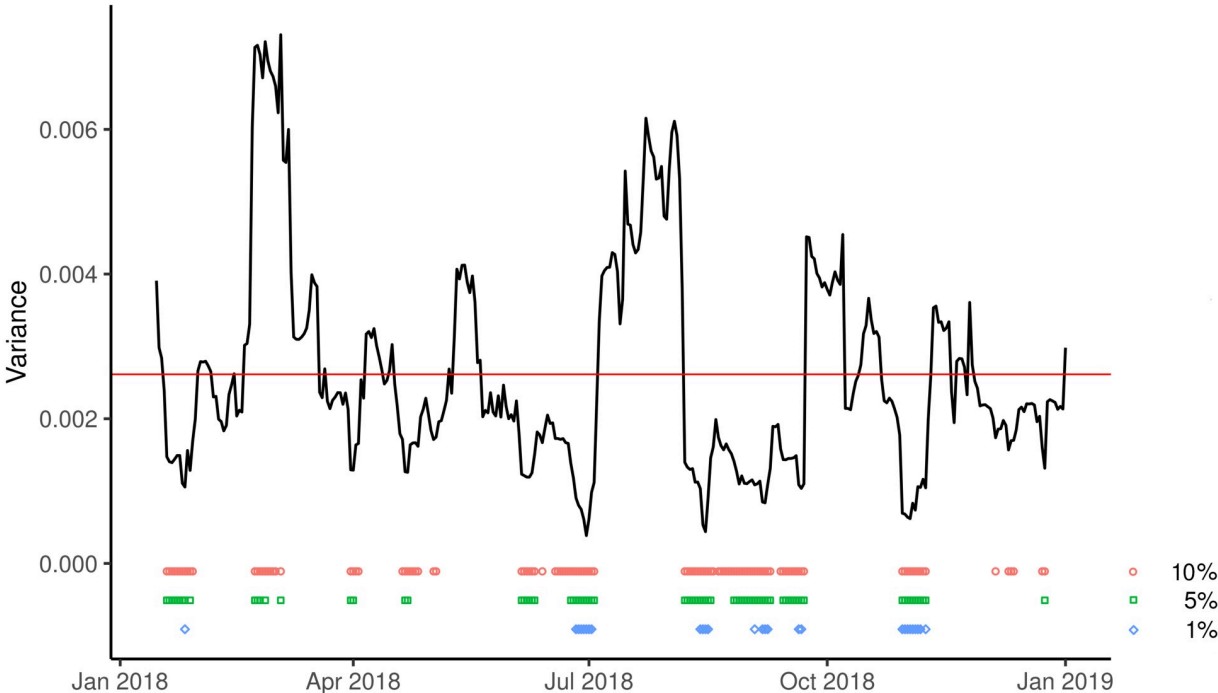

**Fig 4. 15-days running variance of the proportion favourable to vaccination (black line).** Blue circles, green squares, and purple diamonds denote the days when the null hypothesis was rejected at the significance levels of 10%, 5%, and 1%, respectively.

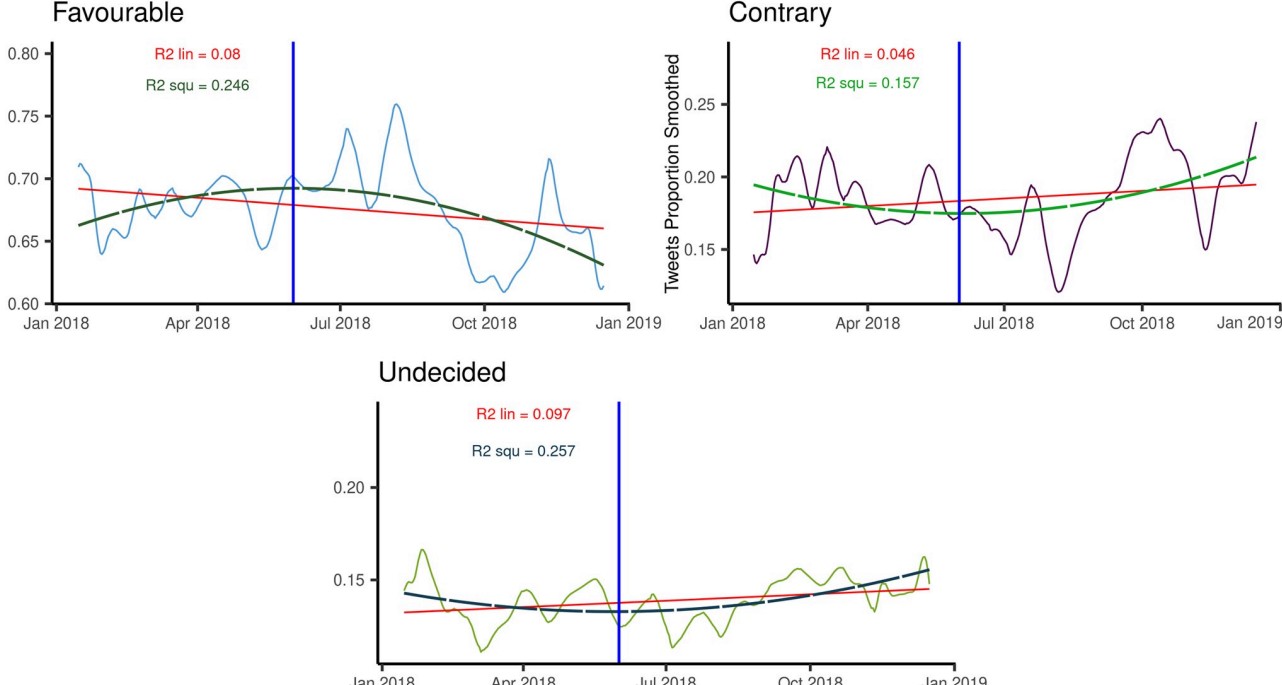

**Fig 5. Kernel smoothing of daily polarity proportion jointly with the corresponding linear and quadratic interpolations.** Panels (a),(b),(c) report the favourable, contrary and undecided proportions, respectively.

government change and the variation in polarity proportions, the association remains of concern in light of its political context.

## Discussion

The contribution's main objective was to investigate whether the 2018 series of contrasting announcements on immunization policy at the highest Italian political level originated disorientation amongst the Italian public. We carried out a sentiment analysis on tweets posted in Italian during 2018 containing vaccine-related keywords.

Our results are as follows. A polarity analysis showed that the proportion of tweets favourable to vaccination was about 70%, the unfavourable one about 16%, while the "undecided" accounted for 13%, in line with similar studies [39–41], yielding an estimate of the hesitant proportion in the range of 30%. As for the temporal trends of tweets, relevant interactions showed clear peaks in correspondence with vaccine-related news and political speeches, indicating that this OSM "is used as an agora for matters of public interest" [42]. Finally, as for the key category of "disorientation", we proposed in this paper a twofold definition namely, short-term disorientation, characterised by unstable, fast-changing, opinions about vaccination, vs long-term disorientation. Our results documented the presence of short-term disorientation by a range of alternative tests. Additionally, a clear yearly trend emerged, showing that the proportion favourable to vaccination increased up to when the previous government–strongly supporting immunization over the media–was in charge (May 2018), and it started declining as soon as the new government, fostering a more ambiguous position, had taken office. We felt hard to believe that this association was unrelated to the new government's continued and ambiguous series of announcements.

Compared to similar studies on vaccination opinions in online social media, we believe that the attempt to define and measure the concept of disorientation and document it is a major strength of the present work. We remark that available data only allowed us to test for disorientation at the "collective" (or aggregate) level. A different research design, tracking users over time (which would require data at the individual level) would in principle allow to investigate disorientation at the individual (or micro-) level.

The reported evidence of disorientation on vaccination is suggestive of the potentially harmful role played by the use of critical health topics for purposes of political consensus. Sadly, these happenings are not new, think, e.g., to the dramatic impact of the denialism of the HIV promoted by a former president of South-Africa in a critical phase of the HIV epidemic [43]. However, these aspects can become especially important due to online social media's increasing role as a source of information (mainly misinformation) [44], which might yield social pressures eventually harmful to vaccine uptake. Said otherwise, persistent disorientation can be inflated by online misinformation, finally drifting into hesitancy. From this viewpoint, we believe that the category of disorientation will deserve future inquiry in more focused studies.

In the Italian case, the effect of disorientation might have been worsened by the almost lack, till the end of 2018, of a stable institutional presence on Twitter by Italian Public Health institutions. This fact, which appears in continuity with the traditional lack of communication between Italian public health institutions and citizens long before the digital era [14], calls for rapid public efforts in terms of an active presence on online social media, aimed to detect and contrast the spread of misinformation and the possible further spread of vaccine hesitancy [45, 46]. Clearly, in the current Italian context, with the ongoing COVID-19 third wave triggered by the new virus variants, the emergency situation has forced a temporary improvement. However, it cannot be disregarded the fact that at the end of the first

pandemic wave (June-July 2020), an amazing large (41%) proportion of Italian adults declared themselves contrary to Covid-19-vaccination (https://www.cattolicanews.it/vaccino-anti-covid-italiani-poco-propensi).

Though not designed for this purpose, this analysis might provide valuable suggestions for vaccine decision-makers. Indeed, the large proportion of hesitant (in the region of 30%) should be carefully considered, if not for their potential impact on current coverage, at least for the social pressure they might enact within online social media, which might eventually feedback negatively on future coverage, as previously pinpointed.

About the limitations of the present analysis, it must be acknowledged that the definitions adopted for the concept of disorientation and its empirical investigation were given in an ad-hoc manner for the present analyses, due to the lack of literature support. From this viewpoint, the category of disorientation will deserve future inquiry in more focused studies, both conceptual and applied. We also have to mention the sub-optimal accuracy of the adopted classification, whose scores are only slightly above those that might result from a random classifier. This was mostly due to the agreement between human annotators, which was somewhat lower than expected. Nonetheless, we feel that these types of issues can often arise when dealing with controversial topics as the one covered by this work, which could trouble also well-trained human annotators.

More in general, the intrinsic limitations of Twitter data (e.g., the maximum length of texts; use of slang, abbreviations, and irony; the tendency to overcome the maximum length by subdividing a single thread into multiple tweets) have largely been acknowledged in the OSM and the related social sciences and public health literature [32, 40].

A further possible limitation lies in the fact that a small proportion of users is highly active and therefore responsible for a substantial proportion of tweets. This can introduce a bias towards the most active users.

From a broader perspective, it must be recalled that the spread of vaccine hesitancy pairs with the widespread diffusion of the so-called "Post Trust Society" [47] and of the "Post Truth Era" [48]. The present investigation can help public health policymakers better orient vaccine-related communication to mitigate the impact of vaccine hesitancy and refusal. This is, however, only a part of the story. Indeed, it is fundamental for public health systems to be able to develop real-time tools to identify fake news as well as tweets hostile to immunization—that might have the largest impact—and appropriately reply to them. This would require that public health communication agencies and institutions are also active in the real-time analysis of online media data, not just in the production of regular communication. On top of this, given the sensible role of the immunization topic, it is surely urgent to develop a moral code preventing the use of such topics for purposes of political consensus and ensuring avoidance of contradictions and ambiguities amongst government members.

A number of previous points might be worth considering in future research, by comparing the language used by tweeters (regardless of their position towards vaccination) and the language of the tweets posted by public health institutions, which represent an important aspect in the communication with agents, particularly with respect to "undecided" individuals, in order to enhance their vaccine confidence. A further point deals with the frequency of fake news spreader users. In this work, we took users as they were, without further control over their profiles. However, this is a key issue deserving careful investigation in future work. Also, the quantitative importance of the followers, which could represent a vehicle for misinformation spreading, possibly distinguished by polarity, as well as that of highly active tweeters, as it emerged in this study, is worth considering in future work on the subject.

## Supporting information

**S1 Fig. The figure represents the real (raw) polarity proportions of favourable (blue), contrary (purple), and undecided (green) with respect to vaccination topic.**
(TIF)

**S1 File. This file contains all the classification reports for all the classifier tested (S1–S5 Tables in S1 File), the keywords adopted to retrieve the tweets (S6 Table in S1 File).**
(DOCX)

**S2 File. This file contains the tweets' Ids containing the Ids and the class of the tweet.**
(ZIP)

## Acknowledgments

We warmly thank three anonymous referees and an Editor of the Journal whose valuable comments allowed us to greatly improve the quality of the manuscript. We also thank Emanuele Del Fava and Alessia Melegaro for their valuable comments on a previous draft of this work.

## Author Contributions

**Conceptualization:** Samantha Ajovalasit, Veronica Maria Dorgali, Angelo Mazza, Alberto d'Onofrio, Piero Manfredi.

**Formal analysis:** Samantha Ajovalasit, Angelo Mazza, Piero Manfredi.

**Methodology:** Samantha Ajovalasit, Angelo Mazza, Piero Manfredi.

**Writing – original draft:** Samantha Ajovalasit, Angelo Mazza, Alberto d'Onofrio, Piero Manfredi.

**Writing – review & editing:** Samantha Ajovalasit.

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
