## [Decision Letter · Decision Letter 0]

22 Apr 2020

PONE-D-20-01302

Evidence of distrust and disorientation towards immunization on online social media after contrasting political communication on vaccines. Results from an analysis of Twitter data in Italy.

PLOS ONE

Dear Ms. Ajovalasit,

Thank you for submitting your manuscript to PLOS ONE. After careful consideration, we feel that it has merit but does not fully meet PLOS ONE’s publication criteria as it currently stands. Therefore, we invite you to submit a revised version of the manuscript that addresses the points raised during the review process.

The reviewers raised major concerns about the manuscript. We invite you to thoroughly review your manuscript by clearly addressing each issue raised by the two reviewers, including the issues about the clarity of manuscript, the methodology used and the soundness of the results and discussion.

We hope that the reviewer reports will allow you to submit a considerably improved version of the manuscript for further consideration.

We would appreciate receiving your revised manuscript by May 29 2020 11:59PM. To enhance the reproducibility of your results, we recommend that if applicable you deposit your laboratory protocols in protocols.io, where a protocol can be assigned its own identifier (DOI) such that it can be cited independently in the future. For instructions see: http://journals.plos.org/plosone/s/submission-guidelines#loc-laboratory-protocols

A rebuttal letter that responds to each point raised by the academic editor and reviewer(s). This letter should be uploaded as separate file and labeled 'Response to Reviewers'.A marked-up copy of your manuscript that highlights changes made to the original version. This file should be uploaded as separate file and labeled 'Revised Manuscript with Track Changes'.An unmarked version of your revised paper without tracked changes. This file should be uploaded as separate file and labeled 'Manuscript',

We look forward to receiving your revised manuscript.

Kind regards,

Alexandre Bovet, Ph.D.

Academic Editor

PLOS ONE

Journal Requirements:

2. In your Methods section, please include additional information about your dataset and ensure that you have included a statement specifying whether the collection method complied with the terms and conditions for the website.

4. Please ensure that you refer to Figure 2 and 3 in your text as, if accepted, production will need this reference to link the reader to the figure.

5. Please include a caption for figure 3.

Reviewers' comments:

Reviewer's Responses to Questions

**Comments to the Author**

1. Is the manuscript technically sound, and do the data support the conclusions?

Reviewer #1: Partly

Reviewer #2: Partly

2. Has the statistical analysis been performed appropriately and rigorously? 

Reviewer #1: Yes

Reviewer #2: I Don't Know

3. Have the authors made all data underlying the findings in their manuscript fully available?

Reviewer #1: No

Reviewer #2: No

4. Is the manuscript presented in an intelligible fashion and written in standard English?

Reviewer #1: No

Reviewer #2: Yes

5. Review Comments to the Author

Reviewer #1: The authors analyzed twitter data involving vaccination-related Italian-language tweets from 2018. They randomly selected 15,000 tweets, which were then manually labelled by 15 students. They found that most of these tweets were composed by “serial twitterers,” with tweets tending to peak around main political events related to vaccination in the Italian context. The majority of these tweets (75%) showed favorable opinion towards vaccination, 14% were undecided, and 11% were unfavorable. The authors argue that there was evidence of “disorientation” among the public.

Overall, the manuscript as it currently stands is difficult to follow. There are many grammatical mistakes throughout the paper, which is distracting. One example comes from the title of a section “Matherials and Methods” (line 97). Many sentences are too long and difficult to follow. The whole manuscript would benefit from a careful reread from the authors and from asking a native English speaker to read over the text to point to language-related issues.

The introduction overall was quite good and provided relevant background for understanding terms related to vaccine hesitancy, vaccination discussions in online formats, and the Italian context. It would have been useful to have more background about which political parties specifically were involved in these political developments in Italy.

The objectives stated in lines 88-93 do not match those provided in the abstract. I have copied and pasted them below. In the abstract, there are 3. In the manuscript, there are 4. These objectives could be tightened up and clarified further. For example, what do the authors mean by “the trend of communication on vaccines on online social media”? This is a broad statement, it is not specific to Italy, and the authors do not consider social media sites outside of Twitter in their analysis. Are authors seeking to establish the prevalence of vaccine hesitancy on Twitter as a proxy for vaccine hesitancy among the actual population in Italy? In my view, these objectives/aims merit further clarification. This would help them better structure the results section.

Objectives and Methods. By a sentiment analysis on tweets posted in Italian during 2018, we attempted at (i) characterising the temporal flow of communication on vaccines over Twitter and underlying triggering events, (ii) evaluating the usefulness of Twitter data for estimating vaccination parameters, and (iii) investigating whether the contrasting announcements at the highest political level might have originated disorientation amongst the public.

(i) describe the trend of communication on vaccines on online social media, (ii) evaluate the potential usefulness of current Twitter data to estimate key epidemiological parameters such as e.g., the hesitant proportion in the population, (iii) evaluating the effectiveness of institutional communication as a tool to contrast misinformation, and (iv) showing evidence that the recent prolonged phase of contrasting announcements at the highest political level on a sensible topic such as mass immunization might have originated a distrust potentially seeding future coverage decline.

I found it difficult to follow the results section because there was not a clear structure in place. It might be helpful for the authors to provide a couple sentences in the introduction and results section that give the reader a sense of knowing what the paper is covering and how it is organized. I was surprised that the concept of “disorientation” was explained in the results section (line 185). If this is an important concept for the authors' analysis, it would have been helpful to have an explanation of it in the introduction.

Some general comments:

I would like to know how the authors determined what was “out of context” (line 148). It would be helpful if the authors provided an example or two.

The authors use the term “serial twitterers.” Would “serial tweeters” be more appropriate? How frequently were these users tweeting? The authors state that they tweet about essentially everything. This is quite vague.

In line 153, it would be more helpful for the reader if the authors state: favorable (F), contrary (c), undecided (U), etc. instead of providing a list of concepts and then using their abbreviations later.

In line 157, the authors’ explanation of “hesitants” left me confused. What are these two sentences about? This merits clarification and more information.

The authors use the term “misinformation” quite a bit. It would be useful to know if they actually examined if the tweets they examined included misinformation. In other words, did they consider that tweets showing unfavorable opinions about vaccination were examples of misinformation?

In line 277, the authors assert that a precondition to establishing trust would be to have more frequent presence of public health authorities in online media. I find such a statement to be quite strong and needs to be backed up with additional data. It might be helpful, but I’m doubtful that the Italian minister of health simply tweeting more about vaccination is a precondition for establishing trust in the public.

Reviewer #2: (also uploaded as PDF)

Review for manuscript "Evidence of distrust and disorientation towards immunization on online social media after contrasting political communication on vaccines. Results from an analysis of Twitter data in Italy."

In this work the authors are analyzing vaccination-related data retrieved from Twitter from 2018 in Italian language and put into the political context during this time. A subset of the data was annotated into 4 categories, those being "favorable", "contrary", "undecided" and "out of context" and a Machine Learning classifier was trained on this data. Predicted data by this classifier was subsequently analyzed, particularly with respect to the absolute counts in each category and their temporal trends. Overall, most tweets were categorized "out of context". Among the relevant category, most tweets were determined to be "favorable" and the rest was subdivided into the categories "contrary" and "undecided". Polynomial fitting was applied to the sentiment trends showing a decline of the "favorable" group towards the end of the year, as well as a slight increase in "contrary" and especially "undecided". The authors then discuss a possible relation between the change of the government to the way vaccination is discussed on Twitter. One of the general conclusions is an increase in "disorientation" due to the ambiguous announcements made by the new government.

The work proposed is interesting and focuses on a relevant topic. However, there is a mismatch between the presented results and the discussion section. The conclusion of there being a direct link between the change of government and the decline in vaccination sentiment and increase in "disorientation" needs to be discussed more clearly. There are several parts of the paper which are unclear and need to be rewritten. I therefore suggest a major revision of this manuscript before publication.

Note that the comments are not given in a specific order. Also, I have not corrected any grammatical mistakes.

Methods

• (minor) The authors mention a total of 4 classes ("favorable", "contrary", "undecided" and "out of context"). It is unclear whether the algorithm was trained on 4 classes or only on 3 classes. If the "out of context" class was simply removed then it means that the predicted data will come from a different underlying distribution than the training data (which could be problematic and should at least be mentioned).

• (minor) Precision, recall and F1 were given for the classifiers. It would be helpful to know the F1 scores for each subclass. Furthermore, it should be mentioned whether these scores are micro or macro averages.

• (minor) Lines 139-144 need better explanation and phrasing. What test was used to determine the degree of freedom for the smoothing? What kernel smoothing procedure?

• (minor) It is not mentioned whether the data was collected through the Twitter API (if so, which endpoint was used?) or via the website. If data was collected via the website it should be written (potentially in the discussion) that the search is not exhaustive and the returned data is filtered by Twitter in terms of relevance/trendingness, which might bias the analysis.

• (minor) It would be very much appreciated if the tweet IDs were published together with the code. This would allow other researchers to reproduce these results. Additionally, given the effort in collecting the annotation data, releasing this data would increase the impact of the work significantly.

Results

• (minor) Figure 1 lacks y-axis labels and legend for the color bar

• (major) It is unclear how the "disorientation" was measured and how it relates to the observed signal. If disorientation is simply a result of the up-and-down trend then one could e.g. plot the variance of the signal over time and see if it increases "sharply" when the government changed. The term "disorientation" is only mentioned in the abstract, title and the beginning of the results section but not in the discussion.

Discussion

• (minor) "After removing noise, the population appeared to be mostly composed by “serial- twitterers” i.e., people tweeting about everything “on top”, including also vaccines, regardless of their awareness of the topic." (Lines 234-236)

What do the authors mean by "serial-twitterers", a group of normal twitter users which also tweet about other things than vaccines? If so, how do the authors know since not all tweets from the timelines of these users were collected? It is also not clear what the term "on top" means in this context. I would recommend to not use the term "serial twitterer" and instead describe this group in another way. Also authors should provide some sort of quantitative reasoning/support for how they allocated users to this group.

• (major) Lines 247-258 discuss how the MMR vaccine coverage relates to the sentiment observed. This should be either moved to the results section or (as the authors state) if not part of the main message of this manuscript it should not be discussed at all. The question of correlation between sentiment and vaccine coverage is an important one, but should be analyzed in more detail and by contrasting e.g. with data from opinion polls before a clear link can be made between Twitter sentiment and vaccination coverage. There is also important literature on this topic which would need to be included in this type of analysis.

"As for the limitations of this work, the main critical point lies in the general relevance of opinion-based information from OSM for predicting trends of vaccine uptake." (Lines 295-296)

The authors mention this as the main limitation of this study. However, as mentioned above vaccine uptake was not properly studied. Therefore, this caveat doesn't apply here.

• (minor) "A key problem is the appropriate modulation of the “language style” to be used by public health communication on online social media." (Lines 280-281)

Since no analysis on language style was performed this should be either left out or rephrased. If kept, authors should include appropriate literature on this topic.

• (minor) "We plan to deep(en) this in future research [...]", (Line 281)

The mentioned research sounds important, but a bit misplaced in the middle of the discussion of the results. Future research should be summarized in a general sense (what is the future research needed to be done by the community as a whole?) at the end and discussed together with caveats.

• (major) "A specific search was therefore carried out over the set of retained tweets by further keywords specifically targeting this situation [...]" (Lines 120-121)

It is unclear which fields of the tweets were searched (user description, text, etc.)? It is also unclear how (if a tweet matched any of the provided keywords) this would directly identify said tweeter as a parent with children in the age of childhood immunization. Later in the discussion it is mentioned that the number of tweets matching the criteria was really small (line 244), therefore it was not analysed further. Although I appreciate the inclusion of negative results, it would be better to move most of it to the results section. Furthermore, as this approach was not successful what was the reason for this? Have the authors tried to expand the search to other keywords? Was the total body of tweets not large enough? The discussion should also involve issues related to identifying demographic subgroups by simple keyword matching (which is obviously problematic).

• (major) "In relation to the growing literature on sentiment analyses and vaccines this is, to the best of our knowledge, the first work on the subject documenting a clear medium-term distrust effect towards immunization arising from persistently ambiguous positions at the highest political level." (lines 291-293)

"Resulting from" is a strong statement, implying direct causation just by observing minor correlations (R2 values are relatively low). This seems to be the main hypothesis of this work but it is not properly discussed. One possible way to discuss causality would be using the Bradford Hill criteria (strength, consistency, temporality, etc.) Some of these criteria might match better, others worse.

• (major) Lines 303-309 are contrasting Twitter to Facebook data and the observation of echo chambers. No Facebook data was analyzed in this study, hence I don't see the need to contrast the collected data with Facebook data. Furthermore, no analysis was conducted with regards to the effects of echo chambers. It is important to address the issues of Twitter data, but it should be limited with respect to the analysis & conclusions in the manuscript.

6. PLOS authors have the option to publish the peer review history of their article (what does this mean?). If published, this will include your full peer review and any attached files.

Reviewer #1: Yes: Michael J. Deml

Reviewer #2: Yes: Martin Müller

---

## [Author Response · Author response to Decision Letter 0]

6 Aug 2020

Response to the comment of reviewer 1. 

• The authors analyzed twitter data involving vaccination-related Italian-language tweets from 2018. They randomly selected 15,000 tweets, which were then manually labelled by 15 students. They found that most of these tweets were composed by "serial twitterers," with tweets tending to peak around main political events related to vaccination in the Italian context. The majority of these tweets (75%) showed favorable opinion towards vaccination, 14% were undecided, and 11% were unfavorable. The authors argue that there was evidence of "disorientation" among the public.

We thank the reviewer for her/his careful reading. We have done our best to improve the paper by strictly following the reviewer points. In this response, we have reported all the reviewer points followed by our responses encapsulated into text boxes. Details on how the referee’s points were incorporated into either the manuscript or the Supplementary Materials are also reported.

• Overall, the manuscript as it currently stands is difficult to follow. There are many grammatical mistakes throughout the paper, which is distracting. One example comes from the title of a section "Matherials and Methods" (line 97). Many sentences are too long and difficult to follow. The whole manuscript would benefit from a careful reread from the authors and from asking a native English speaker to read over the text to point to language-related issues.

Many thanks for the constructive criticism. We have revised the entire manuscript to correct mistakes and to split, or simplify, long or complicated sentences. We have made an effort to sharply improve the English shape. The misprint in the title of the “Materials and Methods” section has been corrected. Additionally, we have attempted to make the titles of the subsections more informative.

The introduction overall was quite good and provided relevant background for understanding terms related to vaccine hesitancy, vaccination discussions in online formats, and the Italian context. It would have been useful to have more background about which political parties specifically were involved in these political developments in Italy.

We thank the reviewer for the appreciation. In the revised draft (Lines 61-65) we have reported a few more details about the political position of the parties involved in the “hot” political debate on vaccination in Italy during 2018 (and related policies).

• The objectives stated in lines 88-93 do not match those provided in the abstract. I have copied and pasted them below. In the abstract, there are 3. In the manuscript, there are 4. These objectives could be tightened up and clarified further. For example, what do the authors mean by "the trend of communication on vaccines on online social media"? This is a broad statement, it is not specific to Italy, and the authors do not consider social media sites outside of Twitter in their analysis. Are authors seeking to establish the prevalence of vaccine hesitancy on Twitter as a proxy for vaccine hesitancy among the actual population in Italy? In my view, these objectives/aims merit further clarification. This would help them better structure the results section.

“Objectives and Methods. By a sentiment analysis on tweets posted in Italian during 2018, we attempted at (i) characterising the temporal flow of communication on vaccines over Twitter and underlying triggering events, (ii) evaluating the usefulness of Twitter data for estimating vaccination parameters, and (iii) investigating whether the contrasting announcements at the highest political level might have originated disorientation amongst the public.

(i) describe the trend of communication on vaccines on online social media, (ii) evaluate the potential usefulness of current Twitter data to estimate key epidemiological parameters such as e.g., the hesitant proportion in the population, (iii) evaluating the effectiveness of institutional communication as a tool to contrast misinformation, and (iv) showing evidence that the recent prolonged phase of contrasting announcements at the highest political level on a sensible topic such as mass immunization might have originated a distrust potentially seeding future coverage decline.”

We apologise for the presence of inconsistencies in the exposition. We have amended both the abstract (Lines >24) and the main text in order to align the number of objectives throughout the entire manuscript (Lines >92) by dropping objective (iii) “evaluating the effectiveness of institutional communication as a tool to contrast misinformation, “ We have also better clarified that Italy during 2018 is the chosen spatio-temporal context of the analysis in the manuscript. In relation to the sentence "the trend of communication on vaccines on online social media", we acknowledge it was vague since the manuscript focused on the “the trend of communication on vaccines on Twitter in Italy during 2018". Consequently, we have rewritten the corresponding sentences in the abstract and introduction."

As for the referee question on whether we were seeking to establish the prevalence of vaccine hesitancy on Twitter as a proxy for vaccine hesitancy among the actual population in Italy, our answer is somewhat articulated. 

As a rule, we can hardly take our twitter evaluations to represent a statistical estimate of the hesitating proportion in Italy. Indeed, as explained in the subsection “True hesitant” of the Results section, the proportion of tweeters actually involved in immunization decision (in an identifiable manner) among the population of people tweeting about the broad subject of vaccines and immunization during 2018 in Italy, was negligible. This was surprising to us if one considers the large age-band involved in compulsory immunizations (from 0 to 15 years of age) and therefore the (possibly large) number of parents’ cohorts involved. This might be due to the fact parents tend to avoid using Twitter to specifically speak about their children. On the other hand, the fact that many people posted tweets with generic contents on immunization is suggestive of the fact that, in the period considered, the topic of immunization had become a topic of general interest amongst the general public opinion, to the point that it was used by politicians for purposes of political consensus rather than aimed at the general interest. 

From this viewpoint, we wanted therefore to suggest that in such situations Twitter might act as a sort of large scale “echo chamber” (ref Cinelli, et al (2020) Echo Chambers on Social media: A comparative Analysis) eventually generating social pressures potentially harmful for vaccine uptake (as reported in the Discussion). 

In the revised Discussion we have made an effort to better expose our viewpoint.

• I found it difficult to follow the results section because there was not a clear structure in place. It might be helpful for the authors to provide a couple sentences in the introduction and results section that give the reader a sense of knowing what the paper is covering and how it is organized. I was surprised that the concept of "disorientation" was explained in the results section (line 185). If this is an important concept for the authors' analysis, it would have been helpful to have an explanation of it in the introduction.

Many thanks for the important point. We have made an effort to improve the structure of the manuscript along the indications of the reviewer. In particular, as suggested, we have made an effort to give more structure to the “Materials and Methods” and “Results” sections. First of all, we have systematically created, in these two sections, parallel subsections dealing with the same topics, which should much improve readability. Moreover, we added a number of sentences to better guide the reader throughout the paper.

We apologise for having been loose in the presentation of the concept of “disorientation” (as also remarked by another referee), which clearly is one of the key concepts of the manuscript, and made an effort to improve the manuscript on this point. 

Motivated by the referee remark, during the revision of the manuscript we have further investigated the literature on social media looking for definitions that could apply to the concept of “disorientation” we had in mind and that we could apply to our investigation, but we weren’t successful. 

Therefore, in the revised “Material and Methods” section we aimed at making our twofold definition of disorientation clearer and have re-organized the manuscript by paying appropriate focus on this concept. In particular, we distinguished carefully between (i) “short-term” disorientation i.e., a state in which people keep changing suddenly and often their opinion on the debated subject possibly as a consequence of being overwhelmed by multiple contrasting information (Lines > 150), and (ii) “longer-term disorientation” (Lines>196) i.e., longer-term trends in opinions caused by persistent ambiguous communication at the highest political level.

In particular, to detect short-term disorientation, we proposed three statistical tests based on the variability of polarity opinions: (i) a general multinomial test, (ii) a running multinomial test, and (iii) a running variance test. These tests are presented at Lines >174 of the revised manuscript. As regards longer-term disorientation, which was already discussed in the original draft, we have made an effort to improve the readability.

Consistently, new results and figures have been added in the Results section to present our new findings on the characterization of short-term disorientation. 

• Some general comments:

• I would like to know how the authors determined what was "out of context" (line 148). It would be helpful if the authors provided an example or two.

In the revised manuscript, in subsection “Data Extraction, transformation and cleaning” (line 106), we have provided a more accurate description of the procedure used to separate the relevant tweets from those we classified as out-of-context. In subsection “Tweets Classification, sentiment analysis, and training set” (line 118) , we added further details on the out-of context category. Finally, in the online appendix, we have report a few – duly anonymised – tweets for each category. 

• The authors use the term "serial twitterers." Would "serial tweeters" be more appropriate? How frequently were these users tweeting? The authors state that they tweet about essentially everything. This is quite vague.

Definitely, the wording "serial tweeters" was more appropriate. However, as also remarked by another referee, in the revised manuscript we preferred to avoid the use of this term due to some possible ambiguities, and decided instead to report a more detailed description of the characteristics of tweeters, which is reported in the subsection “Tweeters” of the Results section. 

First, we have made an analysis on the concentration of the distribution of tweets among tweeters and we found that 1% (5%) of all users tweeted the 30% (50%) of the overall tweet dataset. Moreover, in the online appendix we have reported a Lorenz curve to provide an overall view of the phenomenon.

Finally, to deep the point suggested by the reviewer, we added details on the behaviour of the top 40 users. We found that the majority of them tweeted about everything, especially on debates of a highly polarized nature, most often regardless of having or not an appropriate awareness or background of the topic. Thus, we felt that tweeting about vaccines, seemed to be more representative of this social-hyper activism due to the polarized nature of the subject – rather than by fully awareness of the debate.

• In line 153, it would be more helpful for the reader if the authors state: favorable (F), contrary (c), undecided (U), etc. instead of providing a list of concepts and then using their abbreviations later.

The point was fixed in the subsection “Tweets Classification, sentiment analysis, and training set”. 

• In line 157, the authors' explanation of "hesitants" left me confused. What are these two sentences about? This merits clarification and more information.

We apologies. The original sentence (Line117 of the original draft) was related to the agreed concept of hesitancy in relation to childhood immunization, which was explained at the beginning of the Introduction. This definition clearly applies only to parents of children involved in actual immunization decisions, not to everyone in the general population (and therefore, it possibly applies only to a subset of tweeters). This was explained, but possibly too briefly, at lines 117-120 of the Methods of the original manuscript (and the sentence at line 157 cited by the reviewer was based on the idea of seeking to exactly identify tweets from parents actually involved in vaccination decisions by appropriate keywords). 

We have therefore made an effort to clarify the point by amending both parts of the manuscript. In particular the amended subsection ““True” hesitant parents” should now be much clearer to follow. 

• The authors use the term "misinformation" quite a bit. It would be useful to know if they actually examined if the tweets they examined included misinformation. In other words, did they consider that tweets showing unfavorable opinions about vaccination were examples of misinformation?

This issue is an important one. We apologise if we, unwillingly, abused with theterm “misinformation”. Actually our answer is “no” i.e.,, we deliberately didn’t carry out any analysis aimed to seek whether some tweets (particularly by those unfavourable to vaccination) included misinformation, or were sources of misinformation, because our main goal was to develop a polarity analysis by studying the proportions of opinions (favourable, contrary, undecided) and whether they showed evidence of disorientation according to the definitions provided above (where obviously misinformation can trigger disorientation) regardless of the specific content and of the information sources used (or shared) by tweeters. 

• In line 277, the authors assert that a precondition to establishing trust would be to have more frequent presence of public health authorities in online media. I find such a statement to be quite strong and needs to be backed up with additional data. It might be helpful, but I'm doubtful that the Italian minister of health simply tweeting more about vaccination is a precondition for establishing trust in the public.

Many thanks. The reviewer is correct on the fact that “mere quantitative presence” is not a precondition to generate trust in the public. 

However, in our manuscript we only intended to pinpoint that the traditional lack of adequate communication between Italian public health institutions and citizens, dating back to long before the digital era, seemed to extend until the more recent epoch of diffusion of online social media (as reported in reference 23). This is hardly a productive attitude on a sensible subject as the one of immunization, where national public health institutions should put critical effort to contrast online misinformation and infodemics (see Lachlan et al,2014; If you are quick enough, I will think about it: information speed and trust in public health organizations) triggering hesitancy. Therefore, in the manuscript we only wanted to report simple instances of best practices that public health institution might adopt on online social media.

We have amended the manuscript (Lines 352-361), in order to make our point clear and avoiding misunderstanding in this sense.

• Response to reviewer 2. 

• Review for manuscript "Evidence of distrust and disorientation towards immunization on online social media after contrasting political communication on vaccines. Results from an analysis of Twitter data in Italy."

• In this work the authors are analyzing vaccination-related data retrieved from Twitter from 2018 in Italian language and put into the political context during this time. A subset of the data was annotated into 4 categories, those being "favorable", "contrary", "undecided" and "out of context" and a Machine Learning classifier was trained on this data. Predicted data by this classifier was subsequently analyzed, particularly with respect to the absolute counts in each category and their temporal trends. Overall, most tweets were categorized "out of context". Among the relevant category, most tweets were determined to be "favorable" and the rest was subdivided into the categories "contrary" and "undecided". Polynomial fitting was applied to the sentiment trends showing a decline of the "favorable" group towards the end of the year, as well as a slight increase in "contrary" and especially "undecided". The authors then discuss a possible relation between the change of the government to the way vaccination is discussed on Twitter. One of the general conclusions is an increase in "disorientation" due to the ambiguous announcements made by the new government.

The work proposed is interesting and focuses on a relevant topic.

We thank the reviewer for her/his careful reading and for the appreciation of our work, as well as for the very useful and constructive criticism. We have done our best to improve the paper by strictly following the reviewer points. In this response, we have reported all the reviewer points followed by our responses encapsulated into text boxes. Details on how the referee’s points were incorporated into either the manuscript or the Supplementary Materials are also reported. 

However, there is a mismatch between the presented results and the discussion section. The conclusion of there being a direct link between the change of government and the decline in vaccination sentiment and increase in "disorientation" needs to be discussed more clearly. There are several parts of the paper which are unclear and need to be rewritten. I therefore suggest a major revision of this manuscript before publication. Note that the comments are not given in a specific order. Also, I have not corrected any grammatical mistakes.

In the revised manuscript, we have made a serious effort to clarify the point by rewriting both the Introduction and the Discussion. We have removed sentences suggesting a direct causal link between government changes and the decline in the proportion favourable to vaccination: we clearly wanted to only pinpoint the existence of an association which however deserves to be deepened, in view of its potentially harmful implications for vaccine coverage. We proposed definitions for the concept of “disorientation” (in the Methods section) and improved the related analyses (please, see later responses). Additionally, we have thoroughly revised the entire manuscript and made an effort to increase the clarity of the exposition. 

Methods

• (minor) The authors mention a total of 4 classes ("favorable", "contrary", "undecided" and "out of context"). It is unclear whether the algorithm was trained on 4 classes or only on 3 classes. If the "out of context" class was simply removed then it means that the predicted data will come from a different underlying distribution than the training data (which could be problematic and should at least be mentioned).

We actually trained the algorithm with 4 classes because during early exploratory analyses we soon realised from inspection of tweets that there was a disproportion of tweets belonging to the category that eventually we labelled as "out of context". For simplicity, in the Results section we reported only figures on the three “main” categories F-C-U. In the revised manuscript, we have clarified the point by making several amendments in the subsection “Tweets Classification, sentiment analysis, and training set” of the “Material and Methods” section.

• (minor) Precision, recall and F1 were given for the classifiers. It would be helpful to know the F1 scores for each subclass. Furthermore, it should be mentioned whether these scores are micro or macro averages.

In the online appendix of the revised manuscript we have reported the classification metrics for each classifier. 

We used the 10-fold cross validation on 80% of the labelled sample and we selected the best classifier according to the validation with the remaining 20%. We looked at the classification result where all results of the validation set were in favour of the SVM (see revised appendix). 

Since different classifiers yielded in some cases quite similar results, we used as discriminant between classifiers the F1-weighted score to select the best one. This score calculates first the metrics for each class, and then it does averages by each support (which is the number of true instances per class). Note that “Weighted” alters macro average to account for label imbalance, i.e. calculates the average assigning each class a weight based on the support, this so the F-Score is not between the precision and the recall.

• (minor) Lines 139-144 need better explanation and phrasing. What test was used to determine the degree of freedom for the smoothing? What kernel smoothing procedure?

We used a discrete beta kernel-based smoothing procedure proposed by Mazza and Punzo (2014) in

order to overcome the problem of boundary bias, commonly arising from the use of symmetric kernels. The support of the beta kernel function, in fact, can match our time interval so that, when smoothing is made near boundaries, it allows avoiding the allocation of weight outside the support. The smoothing bandwidth parameter has been chosen using cross-validation.

This part was amended and moved to the subsection on “Longer-term disorientation”.

• (minor) It is not mentioned whether the data was collected through the Twitter API (if so, which endpoint was used?) or via the website. If data was collected via the website it should be written (potentially in the discussion) that the search is not exhaustive and the returned data is filtered by Twitter in terms of relevance/trendingness, which might bias the analysis.

We didn’t use APIs. Data were collected via the website by a scraper using the features of advanced search. Scraping was performed in the best possible manner we could achieve to avoid filtering (in particular, filtering resulting from our accounts). 

In the revised manuscript we have rephrased the corresponding subsection on “Data extraction” in the “Material and Methods” to adequately clarify the point. 

• (minor) It would be very much appreciated if the tweet IDs were published together with the code. This would allow other researchers to reproduce these results. Additionally, given the effort in collecting the annotation data, releasing this data would increase the impact of the work significantly.

That would be an important point. Tweets IDs are now attached as external file provided with the appendix.

To retrieve the original dataset we used the following git repository-user guide. “https://github.com/Jefferson-Henrique/GetOldTweets-python”. 

Tweets were downloaded within a monthly search and then merged to construct the whole dataset, by removing duplicated on ID key a and removed within the non italian tweets with a probabilistic approach.

Results

• (minor) Figure 1 lacks y-axis labels and legend for the color bar

The point has been fixed. In particular, in the revised manuscript we removed the bar.

 • (major) It is unclear how the "disorientation" was measured and how it relates to the observed signal. If disorientation is simply a result of the up-and-down trend then one could e.g. plot the variance of the signal over time and see if it increases "sharply" when the government changed. 

 • (Major)The term "disorientation" is only mentioned in the abstract, title and the beginning of the results section but not in the discussion.

We apologise for having been loose in the presentation of the concept of “disorientation” (as also remarked by another referee), which clearly is one of the key concepts of the manuscript, and made an effort to improve the manuscript on this topic. 

During the revision of the manuscript we have further investigated the literature on social media looking for definitions that could apply to the concept of “disorientation” we had in mind and that we could apply to our investigation but we weren’t successful. 

Therefore, in the revised “Material and Methods” section, we aimed at making our twofold definition of disorientation clearer and have re-organized the manuscript by paying appropriate focus on this concept. In particular, we distinguished carefully between (i) “short-term” disorientation i.e., defined as “a state in which people keep changing suddenly and often their opinion on the debated subject possibly as a consequence of being overwhelmed by multiple contrasting information” (Lines > 150), and (ii) “longer-term disorientation” (Lines>196) i.e., longer-term trends in opinions caused by persistent ambiguous communication at the (highest) political level.

In particular, to detect short-term disorientation, we proposed three statistical tests based on the variability of polarity opinions: (i) a general multinomial test, (ii) a running multinomial test, and (iii) a running variance test, following the reviewer’s indication. These tests are presented at Lines >175 of the revised manuscript. 

As regards longer-term disorientation, which was already discussed in the original draft, we have made an effort to improve the readability of the text.

Consistently, new results and figures have been added in the Results section to present our new findings on the characterization of short-term disorientation. 

Discussion

• (minor) "After removing noise, the population appeared to be mostly composed by "serial- twitterers" i.e., people tweeting about everything "on top", including also vaccines, regardless of their awareness of the topic." (Lines 234-236)

What do the authors mean by "serial-twitterers", a group of normal twitter users which also tweet about other things than vaccines? If so, how do the authors know since not all tweets from the timelines of these users were collected? It is also not clear what the term "on top" means in this context. I would recommend to not use the term "serial twitterer" and instead describe this group in another way. Also authors should provide some sort of quantitative reasoning/support for how they allocated users to this group.

Following the reviewer’s suggestion, in the revised manuscript we preferred to avoid the use of the term “serial tweeters” and decided instead to report a more detailed description of the characteristics of tweeters, which is reported in the new subsection “Tweeters” of the Results section. 

First, we have made an analysis on the concentration of the distribution of the number of tweets by tweeters and we found that 1% (5%) of all users tweeted the 30% (50%) of the overall tweet dataset. Moreover, in the online appendix we have reported a Lorenz curve to provide an overall view of the phenomenon.

Finally, to deep the point suggested by the reviewer, we checked out in detail for the behaviour of the top 40 users. We found that the majority of them tweeted about everything, especially on debates of a highly polarized nature, most often regardless of having or not an appropriate awareness or background of the topic. Thus, we felt that tweeting about vaccines, seemed to be more representative of this social-hyper activism due to the polarization nature of the subject – rather than by fully awareness of the debate.

• (major) Lines 247-258 discuss how the MMR vaccine coverage relates to the sentiment observed. This should be either moved to the results section or (as the authors state) if not part of the main message of this manuscript it should not be discussed at all. The question of correlation between sentiment and vaccine coverage is an important one, but should be analyzed in more detail and by contrasting e.g. with data from opinion polls before a clear link can be made between Twitter sentiment and vaccination coverage. There is also important literature on this topic which would need to be included in this type of analysis.

To avoid ambiguities, we have removed this part.

"As for the limitations of this work, the main critical point lies in the general relevance of opinion-based information from OSM for predicting trends of vaccine uptake." (Lines 295-296)

The authors mention this as the main limitation of this study. However, as mentioned above vaccine uptake was not properly studied. Therefore, this caveat doesn't apply here.

As for the previous point, to avoid ambiguities, we have removed this part.

• (minor) "A key problem is the appropriate modulation of the "language style" to be used by public health communication on online social media." (Lines 280-281)

Since no analysis on language style was performed this should be either left out or rephrased. If kept, authors should include appropriate literature on this topic.

We followed the reviewer suggestion and we left out this part. (line 343)

• (minor) "We plan to deep(en) this in future research [...]", (Line 281)

The mentioned research sounds important, but a bit misplaced in the middle of the discussion of the results. Future research should be summarized in a general sense (what is the future research needed to be done by the community as a whole?) at the end and discussed together with caveats.

We followed the reviewer suggestion and have moved the reported sentence on future research at the end of manuscript. 

• (major) "A specific search was therefore carried out over the set of retained tweets by further keywords specifically targeting this situation [...]" (Lines 120-121)

It is unclear which fields of the tweets were searched (user description, text, etc.)? It is also unclear how (if a tweet matched any of the provided keywords) this would directly identify said tweeter as a parent with children in the age of childhood immunization. Later in the discussion it is mentioned that the number of tweets matching the criteria was really small (line 244), therefore it was not analysed further. Although I appreciate the inclusion of negative results, it (300) would be better to move most of it to the results section. Furthermore, as this approach was not successful what was the reason for this? Have the authors tried to expand the search to other keywords? Was the total body of tweets not large enough? The discussion should also involve issues related to identifying demographic subgroups by simple keyword matching (which is obviously problematic).

We searched the text. 

The reason for trying this approach was that we wanted to investigate the “true” hesitant proportion among tweeters, which seemed to us worth investigating. We failed to do so. We conjectured that this failure depended primarily on the fact that parents tend in general (though not necessarily) to avoid to use Twitter for such purposes.

In the revised manuscript we have made an effort to improve the relevant parts of the Methods, Results, and Discussion to cope with the reviewer’s point.

• (major) "In relation to the growing literature on sentiment analyses and vaccines this is, to the best of our knowledge, the first work on the subject documenting a clear medium-term distrust effect towards immunization arising from persistently ambiguous positions at the highest political level." (lines 291-293)

"Resulting from" is a strong statement, implying direct causation just by observing minor correlations (R2 values are relatively low). This seems to be the main hypothesis of this work but it is not properly discussed. One possible way to discuss causality would be using the Bradford Hill criteria (strength, consistency, temporality, etc.) Some of these criteria might match better, others worse.

About the particular issue of the “low value” of R2, it is to be reminded that we were dealing with strongly oscillating data so that simple trend functions unavoidably yield to low R2 values. For this reason, we found quite encouraging the fact that the R2 figure from the parabolic regression was four times higher than the one from the linear regression, suggesting a strong improvement passing from a linear to parabolic trend. 

As for the parabolic trend, we clearly only observed an association between opinions towards vaccination on Twitter and the political communication provided by the two governments involved during 2018. Therefore, no causation was proved: we apologise for the wrong wording in the original draft, and have carefully modified the corresponding subsection (“Smoothing and longer-term disorientation”) of the Results section along the referee’s indications. 

Thanks for the very important point. 

As for Bradford-Hill causation criteria (many thanks for the suggestion!) we thought that “specificity” and “temporality” (and perhaps, “analogy”) were somewhat supportive of our argument, while other criteria were of more difficult application. Therefore, we preferred to give up.

• (major) Lines 303-309 are contrasting Twitter to Facebook data and the observation of echo chambers. No Facebook data was analyzed in this study, hence I don't see the need to contrast the collected data with Facebook data. Furthermore, no analysis was conducted with regards to the effects of echo chambers. It is important to address the issues of Twitter data, but it should be limited with respect to the analysis & conclusions in the manuscript.

We have removed this part.

---

## [Decision Letter · Decision Letter 1]

16 Sep 2020

PONE-D-20-01302R1

Evidence of disorientation towards immunization on online social media after contrasting political communication on vaccines. Results from an analysis of Twitter data in Italy.

PLOS ONE

Dear Dr. Ajovalasit,

Thank you for submitting your manuscript to PLOS ONE. After careful consideration, we feel that it has merit but does not fully meet PLOS ONE’s publication criteria as it currently stands. Therefore, we invite you to submit a revised version of the manuscript that addresses the points raised during the review process.

In particular, address the issue if the low classification scores shown in Tab 1 of the SI which questions the validity of the results. In a email exchange, reviewer 2 mentionned that he overlooked this issue and wrote "The scores are very low, this should at the very least be mentioned in the caveats. Especially considering that the work builds on the “undecided” category. "

Moreover, the methodology used for collecting, processing and classifying tweets is not explained in sufficient details (see my additional comments below).

Please also address the issues about the clarity of the manuscript raised by Reviewer 1.

We look forward to receiving your revised manuscript.

Kind regards,

Alexandre Bovet, Ph.D.

Academic Editor

PLOS ONE

Additional Editor Comments (if provided):

I thank the authors for having addressed issues raised by the two referees, however there are still important issues with the manuscript, the methodology of the manuscript needs to be better explained and the classification scores are low, which need to be addressed.

Please explain clearly, in order to allow your results to be reproduced, the following points:

- how the Twitter scraper you used works and if there is some rate-limiting,

- what exactly the data filtering and cleaning do,

- how many tweets you collected in total and how many remains after filtering,

- how the smoothing works,

- p.11 line 239, define clearly the "surrounding days",

- what features of the tweets (unigrams, bigrams, trigrams, hasthags, mentions, emojis, ... ?) are used for the classification,

- how the cross-validation is done,

- how "polarity" is defined,

- how the tweet aggregation is done.

In general, please add all the clarifications already asked by the reviewers in the main manuscript.

It is not clear if you are aggregating tweets at the user level or not. If not, this is problematic as you mention that 1% of the users posted 30% of the tweets and therefore your results are strongly biased towards the most active users.

Moreover, since you are interested in users that change their opinions over time (disorientation), you could track specific users and measure how the opinions of their tweets change over time this would help you to validate your measure of disorientation.

Please remind the readers of what is the null hypothesis in the figure captions and clearly define each plot lines (blue, green, purple).

Please report the average training scores in the main manuscript.

The training scores are low, in particular for the undecided class upon which the results are built. Please comment on the validity of the results. Could you improve the classification by using a different set of features?

Reviewers' comments:

Reviewer's Responses to Questions

**Comments to the Author**

1. If the authors have adequately addressed your comments raised in a previous round of review and you feel that this manuscript is now acceptable for publication, you may indicate that here to bypass the “Comments to the Author” section, enter your conflict of interest statement in the “Confidential to Editor” section, and submit your "Accept" recommendation.

Reviewer #1: (No Response)

Reviewer #2: All comments have been addressed

2. Is the manuscript technically sound, and do the data support the conclusions?

Reviewer #1: Partly

Reviewer #2: Yes

3. Has the statistical analysis been performed appropriately and rigorously? 

Reviewer #1: Yes

Reviewer #2: Yes

4. Have the authors made all data underlying the findings in their manuscript fully available?

Reviewer #1: No

Reviewer #2: Yes

5. Is the manuscript presented in an intelligible fashion and written in standard English?

Reviewer #1: No

Reviewer #2: Yes

6. Review Comments to the Author

Reviewer #1: The authors have addressed many of my initial concerns from the first round of reviews. Regarding the scientific rigor of the paper, the authors have provided interesting results. That said, the presentation of the manuscript continues to be difficult to follow for me. The paper includes many aspects that are not well linked together, in my view. The paper is very busy in the sense that it introduces many different items and does not adequately pull them together to give the reader a sense of why what they did is important.

If the authors wish to make a major point of the paper to define the concept of disorientation, as they state on p. 16, line 328, then I would expect this to be a clearer issue in the introduction section of the paper. If disorientation is an issue that the authors want to better define, the introduction of this concept and related literature should be presented in the introduction section, and not in the methods section (lines 145 and on, where it is first defined).

That said, I do not fully understand why the concept is introduced in the first place. On line 145, the authors state, "To the best of our knowledge, the concept of "disorientation" does not seem to have been well defined in the literature of online social media. Properly defining the concept of "disorientation" can be complicated, e.g., it can be simply a consequence of the lack of adequate information, but also of the over-exposition to information including misinformation."

If the topic has not been well-defined in this literature, it would be helpful for the reader to know if the term has been used at all, and in what papers/articles. To me, this was difficult to read because it sounds like the authors have decided at this point that disorientation was a concept they were interested in, it has not been covered in the literature, but they are going to use it anyways. This is not a problem, per se, but it could be presented in a much easier to follow and coherent fashion.

The authors mentioned having addressed language-related issues and long sentences throughout the paper, but I was able to identify language issues already in the abstract. Lines 25 - 28, "attempted at (i) characterizing...(ii)evaluating..." etc. This should be "attempted TO (i) characterizE...(ii) evaluatE..." etc. Line 37, "critical health topics, as immunization"  this sentence should include "such" between "topics," and "as." There were other language related issues throughout the manuscript. Line 61: oppositions  opponents. This sentence is also very long. Line 76. "troughs"  "through." There were additional grammatical issues, but I have not outlined them all here. The authors again used many long sentences with multiple subordinate clauses. For example, the first paragraph of the discussion section is composed of 2 very long, confusing sentences (Lines 285-292).

The graphics could also be better explained with legends for the colors.

For readability, it would be helpful if the authors took a line-by-line reading to clarify all sentences and shorten them to make the paper easier to follow for the reader.

Reviewer #2: The work has now greatly been improved and all comments have been addressed. A minor comment: Authors may want to increase DPI on the figures (and if jpg was used to use the PNG format instead), in order to avoid blurriness.

7. PLOS authors have the option to publish the peer review history of their article (what does this mean?). If published, this will include your full peer review and any attached files.

Reviewer #1: No

Reviewer #2: **Yes: **Martin Müller

---

## [Author Response · Author response to Decision Letter 1]

16 Mar 2021

#Reviewer 1 

1 The authors have addressed many of my initial concerns from the first round of reviews. Regarding the scientific rigor of the paper, the authors have provided interesting results. That said, the presentation of the manuscript continues to be difficult to follow for me. The paper includes many aspects that are not well linked together, in my view. The paper is very busy in the sense that it introduces many different items and does not adequately pull them together to give the reader a sense of why what they did is important.

1A We thank the reviewer for her/his careful reading of the revised draft, and we apologize for having failed in achieving adequate clarity. In this further revised version of the manuscript, we have done our best to improve the paper by strictly following the reviewer's suggestions in her/his second report. In this response, we have reported all the reviewer points followed by our responses encapsulated into text boxes. Details on how the referee's points were incorporated into either the manuscript or the Supplementary Materials are also reported. The notation "see LXXX" specifies the line of the revised manuscript where the related information can be found.

2 If the authors wish to make a major point of the paper to define the concept of disorientation, as they state on p. 16, line 328, then I would expect this to be a clearer issue in the introduction section of the paper. If disorientation is an issue that the authors want to better define, the Introduction of this concept and related literature should be presented in the introduction section, and not in the methods section (lines 145 and on, where it is first defined).

That said, I do not fully understand why the concept is introduced in the first place. On line 145, the authors state, "To the best of our knowledge, the concept of "disorientation" does not seem to have been well defined in the literature of online social media. Properly defining the concept of "disorientation" can be complicated, e.g., it can be simply a consequence of the lack of adequate information, but also of the over-exposition to information including misinformation."

If the topic has not been well-defined in this literature, it would be helpful for the reader to know if the term has been used at all, and in what papers/articles. To me, this was difficult to read because it sounds like the authors have decided at this point that disorientation was a concept they were interested in, it has not been covered in the literature, but they are going to use it anyways. This is not a problem, per se, but it could be presented in a much easier to follow and coherent fashion.

2A Yes, we definitely believe that the definition of disorientation and its measurement /testing using data on the temporal path of polarity proportions was the key contribution of this work. We, therefore, thank the reviewer for her/his further suggestions about the presentation of the topic (and apologize for the "bad selling" of our idea). In the revised draft, we carefully followed the reviewer's suggestions, in particular:

 • We introduced our key ideas about "disorientation" in the Introduction (moving there a part of the contents originally presented in the Methods, suitably revised). Therein we also motivate our choice to introduce a new definition in view of the (to the best of our search on the literature) lack of equivalent concepts in the literature on OSM and other relevant literatures. See main text from L-82

 • We left in the "Materials and Methods" the technical part on the statistical methodology used to document the presence of disorientation through appropriate tests on time series of polarity proportions. See main text from L-159

3 The authors mentioned having addressed language-related issues and long sentences throughout the paper, but I was able to identify language issues already in the abstract. Lines 25 - 28, "attempted at (i) characterizing...(ii)evaluating..." etc. This should be "attempted TO (i) characterizE...(ii) evaluatE..." etc. Line 37, "critical health topics, as immunization"  this sentence should include "such" between "topics," and "as." There were other language related issues throughout the manuscript. Line 61: oppositions  opponents. This sentence is also very long. Line 76. "troughs"  "through." There were additional grammatical issues, but I have not outlined them all here. The authors again used many long sentences with multiple subordinate clauses. For example, the first paragraph of the discussion section is composed of 2 very long, confusing sentences (Lines 285-292).

3A We did our best to revise the new version of the manuscript thoroughly. Additionally, all the points raised by the reviewer have been carefully fixed.

4 The graphics could also be better explained with legends for the colors.

4A We added legends for the colours in all the figures using different colours.

5 For readability, it would be helpful if the authors took a line-by-line reading to clarify all sentences and shorten them to make the paper easier to follow for the reader.

5A We did our best to clarify sentences and improve the overall readability.

## Reviewer 2 

The work has now greatly been improved and all comments have been addressed.

We thank the reviewer for her/his appreciation of our effort in responding to the reviewer's comments. Below you find our response to your last point. 

Additional to this, the Editor mentioned an email exchange where you suggested that "The scores are very low, this should at the very least be mentioned in the caveats. Especially considering that the work builds on the "undecided" category. "

In order to respond to this point, during the preparation of this revision of the manuscript, we have critically re-analyzed the entire research path and carefully re-checked all steps of data preparation and analysis. This permitted us to correct some problems in the data preparation, which allowed us to completely solve the problem of the "low classification scores shown in Tab 1 of the SI", consequently achieving reasonable values for all classes considered. This can be appreciated from a new Table (Table 1) that has been added to the main text (upon request by the Editor) reporting the involved training scores, including average training scores, as requested. 

Entering into details, the detected problem in classification was responsible for the excessively large proportion of out-of-context tweets over total data. This, in turn, was yielding to low numbers of tweets for the two less represented categories ("contrary" and "undecided"), mechanically leading to low precision, which is a problem well acknowledged in the literature. 

The discovery of the bug allowed to solve all these problems altogether, eventually providing very reasonable results.

1 A minor comment: Authors may want to increase DPI on the figures (and if jpg was used to use the PNG format instead), in order to avoid blurriness.

1A The point has been fixed. High-quality figures were provided by substantially expanding the number of DPI.

---

## [Decision Letter · Decision Letter 2]

9 Jun 2021

Evidence of disorientation towards immunization on online social media after contrasting political communication on vaccines. Results from an analysis of Twitter data in Italy.

PONE-D-20-01302R2

Dear Dr. AJOVALASIT,

We’re pleased to inform you that your manuscript has been judged scientifically suitable for publication and will be formally accepted for publication once it meets all outstanding technical requirements.

Kind regards,

Alexandre Bovet, Ph.D.

Academic Editor

PLOS ONE

Additional Editor Comments:

Please take into account the issues raised by of the Reviewer when preparing the final version.

Please also mention the imperfectness of the classification as a limitation of the results in the discussion. The classification scores are only slightly above the ones of a random classifier.

Reviewers' comments:

Reviewer's Responses to Questions

**Comments to the Author**

1. If the authors have adequately addressed your comments raised in a previous round of review and you feel that this manuscript is now acceptable for publication, you may indicate that here to bypass the “Comments to the Author” section, enter your conflict of interest statement in the “Confidential to Editor” section, and submit your "Accept" recommendation.

Reviewer #3: (No Response)

2. Is the manuscript technically sound, and do the data support the conclusions?

Reviewer #3: Yes

3. Has the statistical analysis been performed appropriately and rigorously? 

Reviewer #3: Yes

4. Have the authors made all data underlying the findings in their manuscript fully available?

Reviewer #3: Yes

5. Is the manuscript presented in an intelligible fashion and written in standard English?

Reviewer #3: Yes

6. Review Comments to the Author

Reviewer #3: In this manuscript, the authors analyze the polarity of vaccine relevant tweets during a time period in which there were multiple changes in vaccine policy in Italy that may have shifted opinions on vaccination. Overall, the statistical analysis is clearly described and the results are interesting and relevant.

My only concern is minor and is with the introduction of the concept of disorientation – given the title of the paper I expected the focus to be on disorientation and yet in the abstract it is only briefly mentioned as one of three objectives and there is no clear definition of disorientation (which is not a concept that I was familiar with prior to this manuscript and many readers may not be familiar with). I would suggest adding definitions of disorientation to the abstract to orient readers to the concept as they start reading the manuscript. Second, I would suggest re-emphasizing the definitions of short- and long-term disorientation defined in the introduction in the methods section describing how short and long-term disorientation were detected in the data (starting at p.8, line 159).

7. PLOS authors have the option to publish the peer review history of their article (what does this mean?). If published, this will include your full peer review and any attached files.

Reviewer #3: No

---

## [Editor Report · Acceptance letter]

25 Jun 2021

PONE-D-20-01302R2 

Evidence of disorientation towards immunization on online social media after contrasting political communication on vaccines. Results from an analysis of Twitter data in Italy. 

Dear Dr. AJOVALASIT:

I'm pleased to inform you that your manuscript has been deemed suitable for publication in PLOS ONE. Congratulations! Your manuscript is now with our production department. 

Kind regards, 

on behalf of

Dr. Alexandre Bovet 

Academic Editor

PLOS ONE